# SQL-GEN: Bridging the Dialect Gap for Text-to-SQL Via Multi-Dialect Synthetic Data And Model Merging

## Abstract

Text-to-SQL systems, that convert natural language queries into SQL programs, have seen significant progress with recent breakthroughs. However, these have been primarily for the SQLite dialect and adapting Text-to-SQL systems to other SQL dialects like BigQuery and PostgreSQL remains a challenge due to the diversity in SQL syntaxes and functions, along with the high cost of collecting and curating SQL-specific training data. To this end, we introduce SQL-GEN, a framework for generating high-quality synthetic data for any dialect guided by dialect-specific tutorials. We demonstrate the effectiveness of SQL-GEN in creating training data to significantly improve the downstream Text-to-SQL performance for other dialects – it improves the execution accuracy by up to 20% over previous methods, and reduces the gap with large-scale human-annotated data on unseen real world multi-dialect benchmarks. Moreover, combining our synthetic data with human-annotated data provides additional performance boosts up to 5.6%. Towards unifying the multi-dialect capability in a single system, we also introduce a novel Mixture of Experts (MoE) initialization method that integrates dialect-specific models by merging self-attention layers and initializing the gates with dialect-specific keywords, yielding one unified and versatile model adept for multiple SQL dialects, further enhancing performance across different SQL dialects. By leveraging shared core features of multiple dialect-specific models, our MOE demonstrated superior performance compared with models trained on individual dialects alone.

## 1 Introduction

Text-to-SQL systems translate natural language questions into executable SQL queries, enabling users to interact with databases using natural language. This transformation is crucial as it links the intuitive nature of human communication with the structured precision of SQL, the standard language for querying databases (Androutsopoulos et al., 1995; Hristidis et al., 2003; Li & Jagadish, 2014). Text-to-SQL plays a significant role for conversational agents, empowering them to process complex queries within large-scale databases efficiently (Yu et al., 2019; Gu et al., 2022; Pérez-Mercado et al., 2023). Such systems serve as a copilot for data science professionals to enhance productivity, beyond being valuable for non-technical users who wish to derive business insights without SQL expertise (Li et al., 2023; Sun et al., 2023a;b; Wang et al., 2019).

SQL has been adopted by each database product (e.g. PostgreSQL, MySQL, and SQLite) to suit their specific needs. Despite their common foundations, these SQL dialects differ significantly in their syntax, functions, and capabilities, which even make the automated translation of queries across dialects a complex task that often requires human intervention (Zmigrod et al., 2024; Ngom & Kraska, 2024). Figure 1 exemplifies a question that can be answered with different SQL keywords across different dialects with their own unique keywords that are distinct from one another. Additionally in Appendix A.7, we provide some of the dialect specific keywords for BigQuery, PostgreSQL, and SQLite, which are not supported across all of them. In the realm of Text-to-SQL, most benchmarks are based on the SQLite dialect, chosen for its simplicity and self-contained nature (Li et al., 2024b; Yu et al., 2018b; Zhong et al., 2017; Chang et al., 2023; Gan et al., 2021). This dialect dependency poses a significant challenge, as models trained on SQLite-specific syntax are prone to generating

erroneous queries in other dialects. A conventional solution involves translating queries across dialects before training, using tools like SQLglot parser or tools offered by cloud providers (Li et al., 2024b; Mao, 2023; Zmigrod et al., 2024). However, most of these tools are not 100% successful in translating the queries between dialects Zmigrod et al. (2024). For example, during the translation of the BIRD benchmark (Li et al., 2024b) from SQLite to BigQuery, approximately 20% of the queries encountered errors using the SQLGlot parser. Additionally, this approach fails to leverage the unique capabilities of each SQL dialect, as queries originally written for SQLite may not fully exploit the potential of the target dialects due to the absence of support for specific functions and keywords in the source dialect. For example, REGEX operations are supported in BigQuery but not in SQLite, so we cannot get this REGEX support by translating queries from SQLite. To overcome this, we propose a dialect-agnostic method for generating synthetic Text-to-SQL pairs for any database.

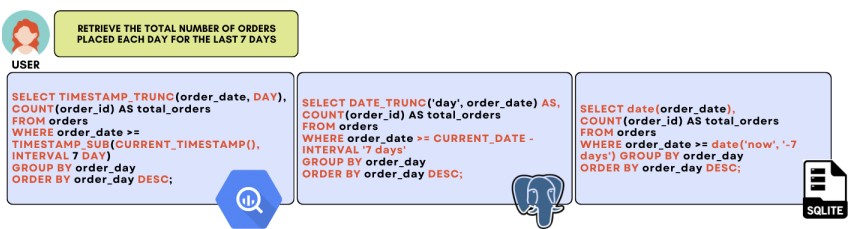

Figure 1: Exemplification of a question being answered using different SQL keywords for different dialects, BigQuery, PostgreSQL, and SQLite.

SQL-GEN consists of a three-step pipeline. Initially, we begin with a small collection of seed SQL templates. These retain only the SQL keywords, abstracting away the specific database schema and values. Accompanying these templates, we provide dialect-specific tutorials that explain the usage of each SQL keyword across different dialects. In the first stage of our pipeline, we leverage a Large Language Model (LLM) to expand the seed templates, using the tutorials to adapt the keywords and function to various SQL dialects. In the second stage, these database-independent templates are populated with actual values and schema elements from any given database. The final stage involves a rigorous quality checks to ensure that the generated pairs accurately match each other.

We perform comprehensive evaluations on the effectiveness of SQL-GEN in teaching models new dialects, specifically focusing on dialect-specific characteristics like keywords. We assess the quality of synthetic samples by comparing them against both prior synthetic and human-annotated datasets. We construct synthetic datasets for three dialects and evaluate various LLMs (of sizes 7B-22B), trained on these pairs as well as other baselines. We show that all LLMs trained on our synthetic data exhibit a performance increase ranging from 4% to 27%, surpassing those trained on earlier synthetic and human-annotated data. In addition, for under-explored dialects of PostgreSQL and BigQuery, we focus on evaluations on real-world data, specifically designed for PostgreSQL and BigQuery. We demonstrate that models trained with synthetic data generated by SQL-GEN consistently outperform others by a significant margin, approximately 7.5% on BigQuery and 2.5% on PostgreSQL dialect-specific datasets. This highlights the generalizability of our approach to unseen datasets due to its broad coverage. We also explore data augmentation for cross-domain Text-to-SQL scenario as another use case of synthetic queries. By integrating synthetic queries with training samples from other databases, we show improvements in models' ability to adapt across domains. We test the proposed data augmentation approach using the BIRD development set by combining synthetic and training data, aiming to improve performance on the target databases. We show consistent performance improvements of up to ∼5.7% when fine-tuning with the augmented data.

As noted earlier, each SQL dialect has distinct keywords and functions, rendering a model trained on a specific dialect uniquely specialized. A significant challenge arises when Text-to-SQL users manage databases across multiple dialects, as deploying multiple dialect-specific models can be computationally demanding. Moreover, we hypothesize that multi-dialect datasets share common SQL features, leading to some overlap in the features learned by the models. By merging models, we believe they can gain a deeper understanding of core features as they appear across multiple dialects. To overcome this, we introduce a novel method for utilizing the Mixture of Experts (MoE) architecture (Fedus et al., 2022b; Riquelme et al., 2021; Jiang et al., 2024). Specifically, our approach is based on initializing the MoE model using the layers of the dialect-expert models, while the sub-layers are initialized using a two-by-two Spherical Linear Interpolation (SLERP) of self-

attentions from the dialect experts, as approaches for efficient merging. Additionally, to harness dialect-specific expertise effectively, we initialize the routers with hidden states corresponding to dialect-specific keywords. We demonstrate an improvement of 2.5% in average performance compared to other model merging approaches, as well as superior performance in SQLite and BigQuery dialects, outperforming the expert models by 0.68% and 7.25%, respectively. We also show performance outperformance of a MoE model initialized without our dialect-aware model merging, trained for the same number of steps.

## 2 METHODOLOGY

We first introduce the SQL-GEN pipeline, designed to generate high-quality, dialect-specific Text-to-SQL samples, as illustrated in Figure 2 and detailed in Algorithm 1. The generation of multi-dialect Text-to-SQL synthetic data addresses the critical issue of insufficient high-quality data for training models tailored to specific SQL dialects. However, since users of Text-to-SQL systems often work with databases across various dialects, serving models in a multi-dialect environment presents a unique challenge. Additionally, many SQL dialects adhere to standard SQL and share common syntaxes, making the concept of information sharing particularly compelling. To this end, we propose a model merging method that combines dialect-specific models into a single, unified MoE model capable of serving multiple SQL dialects effectively.

---

**Algorithm 1** SQL-GEN: Dialect-Specific Synthetic Question-SQL Pair Generation.

---

**Require:** set of target databases $D$, set of dialect-specific tutorials for each SQL keyword $T$, LLM $M_1$, quality assurance LLM $M_2$, number of SQL templates threshold $\theta$, number of question-SQL pairs threshold $\beta$, template expansion prompt $P_{TempGen}$ Figure 9, question-SQL sample generation prompt $P_{Gen}$ Figure 10, quality assurance prompt $P_{Quality}$ Figure 12, SQL templates parsing filter $Filter_1(.)$, question-SQL pairs heuristic-based filters $Filter_2(.)$, template extractor function $G(.)$

1: **Initialization:**
2: $S \leftarrow G(SeedQueries)$ *// Generate initial set of SQL templates using a set of dialect-specific seeds*
3: $T \leftarrow Dialect\ Specific\ Tutorials$ *// Scrape a set of dialect-specific SQL tutorials*
4: **while** len$(S) < \theta$ **do**
5:     *// Generating new SQL templates using tutorials*
6:     $template \leftarrow$ sample$(S)$
7:     $tutorial \leftarrow$ sample$(T)$
8:     $S \leftarrow S \cup \{Filter_1(M_1(P_{TempGen}(template, tutorial)))\}$
9: **end while**
10: $Q \leftarrow \emptyset$ *// set of generated question-SQL pairs*
11: **while** len$(Q) < \beta$ **do**
12:     *// Generating question-SQL pairs*
13:     $template \leftarrow$ sample$(S)$
14:     $db \leftarrow$ sample$(D)$
15:     $candidate \leftarrow Filter_2(M_1(P_{Gen}(template, db)))$
16:     **try**
17:         $results \leftarrow$ executeQuery$(candidate)$ *// quality assurance check*
18:         **if** isError$(results)$ **then**
19:             **continue** *// Skip to the next iteration if error*
20:         **end if**
21:     $Q \leftarrow Q \cup \{Filter_2(M_2(P_{Quality}(candidate, results)))\}$
22: **end while**

---

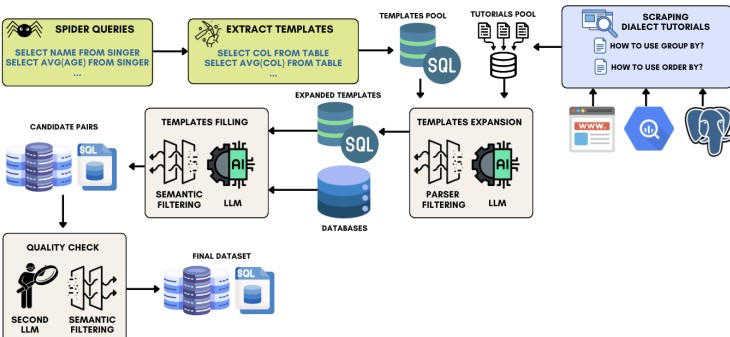

Figure 2: SQL-GEN to generate diverse and high-quality synthetic Text-to-SQL samples for any database.

## 2.1 Synthetic Text-to-SQL Data Generation

The initial step of SQL-GEN involves creating a pool of simple queries by extracting template question-SQL pairs. Building on this, we expand the templates using LLMs and dialect-specific tutorials, rather than relying solely on extracted templates. After expanding these templates, each one is converted into an actual SQL query, and a corresponding question is generated by passing a sample database to the LLM. Subsequently, all generated question-SQL pairs, along with their execution results, undergo a quality-checking step to ensure they accurately match each other and effectively extract valuable information from the database. Throughout this process, we apply filtering to remove low-quality samples at each step, to ensure the overall generated data would be high quality.

**Extraction of Seed Templates:** Similar to Wu et al. (2021); Yu et al. (2020), we extract SQL templates by abstracting all of the schema mentions in the queries from the Spider dataset to serve as a foundational pool for generating more diverse queries. Since the seed queries are initially in SQLite, for the other dialects, we transpile these queries using the SQLGlot parser (Mao, 2023) before extracting their SQL query templates.

**Templates Expansion From Tutorials:** The initial pool of query templates created in the previous step presents two main challenges. First, the extracted templates are derived from simple SQL queries, which are relatively basic compared to the queries found in other SQLite benchmarks like BIRD (Li et al., 2024b). Second, for dialects other than SQLite, the seed templates—originally designed for SQLite would not have complete coverage for all the dialect-specific SQL functions from other dialects. To address these, we expand the templates for each dialect using LLMs with in-context learning (Brown et al., 2020; Wei et al., 2022). To prepare the LLMs for template expansion, we first scrape online tutorials for each target dialect, focusing on the use of dialect-specific SQL functions and keywords. We then randomly select a seed template from the pool, pairing it with a random tutorial document about a dialect-specific keyword or function, and prompt the LLM to increase the complexity of the template, drawing inspiration from the document. To ensure the validity of the templates for all of the different dialects, we parse all generated SQL templates using dialect-specific parser (from SQLglot (Mao, 2023)). Figure 3 demonstrates an examples of this template expansion step for the BigQuery dialect. Additionally, Appendix A.9.1 provides the prompt that has been used for template expansion.

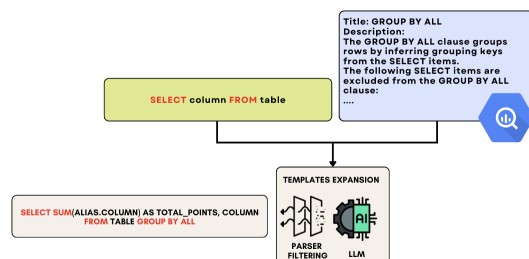

Figure 3: An example of template expansion using BigQuery tutorials and seed templates.

**Sample Generation:** After generating the SQL templates, our next step is to convert them into valid question-SQL pairs. For this process, we select a template along with a database schema with a random row from any given database. Random database rows are necessary since the LLM should be able to fill conditions with actual database values. The database schemas can be sourced from different datasets (e.g. publicly-available Spider or BIRD). As these are originally in SQLite, these databases are migrated to each target dialect for dialects other than SQLite. The combination is then passed to an LLM, instructing it to integrate schema mentions into the templates and generate corresponding questions that align with the SQL queries. After generating the SQL queries, heuristic-based semantic and syntactic filters are applied to ensure the high quality of both the queries and questions. The specifics of these filters are detailed in the Appendix A.6. Additionally, Appendix A.9.2 includes the detailed prompt which is used in this step.

**Data Quality Check:** To ensure high quality generation of question-SQL pairs, we present the question-SQL pairs alongside the first $K$ rows of their execution results over the database to an LLM. This LLM is tasked with verifying that the question and SQL pair match appropriately and that the question is free of ambiguity. To avoid repeating the same errors, we employ a different LLM,

not used in previous steps, to act as the judge. Appendix A.4.1 provides a detailed analysis of the importance of utilizing a secondary LLM and highlights the importance of this step. Appendix A.9.3 provides the prompt that has been used for quality checking.

## 2.2 DIALECT EXPERTS MERGING: MULTI-DIALECTS MIXTURE OF EXPERT (MoE)

With SQL-GEN, we can generate question-SQL pairs for various dialects and train corresponding dialect-specific models. However, in real-world scenarios, users often manage databases across different dialects, necessitating the deployment of multiple models, which can come with practical challenges, including increased model serving costs and overhead of managing multiple checkpoints. Additionally, while each dialect features unique keywords and functions, there is commonality across some SQL keywords across dialects that can be exploited for cross-dialect information transfer. By merging these dialect experts into a single model, not only we mitigate the practical serving challenges, but we can improve the performance of each facilitating sharing of common knowledge. As model merging approaches, we introduce our proposed method utilizing the Mixture of Experts (MoE) architecture.

**MoE:** In a Mixture of Experts (MoE) model, each layer contains multiple MLP blocks, or "experts," and a router network selects specific experts to process each token at every position, combining their outputs. This architecture enhances the traditional MLP sub-layer within Transformer blocks by replacing it with multiple experts, each with its own set of parameters (Jiang et al., 2024; Fedus et al., 2022a). MoE-based LLMs route tokens to different experts, increasing modeling expressiveness without significantly increasing the compute budget, as only a subset of experts is activated for each token. With different Transformer-based expert models, we can combine them into a single MoE model that leverages the expert-specific MLP layers. By initializing the router to select the corresponding expert for each token, we can combine the knowledge of expert models by activating multiple experts and merging the self-attention layers. This approach aligns well with our setting, where we have dialect-specific expert models that already have prior knowledge of SQL syntax. For dialect-specific keywords, we can use the router to select the appropriate MLP layer, which allows us to integrate three models into a single one without the need to train a new model from scratch. Figure 4 illustrates one Transformer block with the proposed method for constructing an MoE model from three distinct dialect expert models.

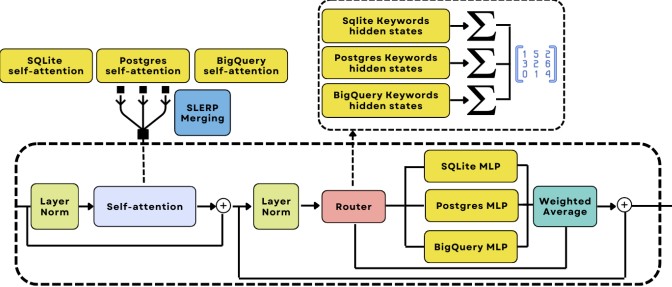

Figure 4: Our proposed method to initialize one Transformer block of a MoE model from different dialect experts, exemplified here for Postgres, SQLite, and BigQuery dialects to create an all in one model to address all. Objects in yellow demonstrate multi-dialect models

**Key-words based multi-dialects gating (routing):** An important aspect of the MoE framework is the gating (routing) mechanism. In MoE, the output for a given input $x$ is computed as a weighted sum of the expert networks' outputs, with the weights determined by the gating network (Jiang et al., 2024). Given n expert networks $\{E_1, E_i, ..., E_n\}$ The output is

$$\sum_{i=0}^{n-1} G(x)_i E_i(x)$$

where the gates' outputs are determined based on the dot product of the input $x$ and the gate weights $W_g$ as follows:

$$G(x) = Softmax(TopK(dot(x, W_g)))$$

We propose to initialize these gates at each layer by averaging the hidden vectors of the dialect-specific keywords, derived from the training data of each model and based on the top $K$ most frequently occurring dialect-specific keywords from generated question-SQL pairs. This process begins by cataloging all dialect-specific keywords from our generated SQL queries, sorting them by frequency, and selecting the top-k keywords. These keywords are then processed by the model, where the hidden representations from the self-attention sub-modules for all tokens of these keywords are used to initialize the gates. The formula below provides the described method for initializing the gate weights:

$$W_g(i) = \frac{1}{K} \sum_{k=1}^{K} \sum_{j=1}^{k_t} h_{k_j}$$

, where $W_g(i)$ is the ith column of the gate weight matrix corresponding to ith expert, $k_t$ is the number of tokens for the kth dialect keyword, and $h_{k_j}$ is the hidden representation of the j-th token of the k-th keyword. This approach increases the dot product between dialect-specific input keywords and their corresponding gate weight matrix columns, thereby boosting the weight for the dialect-specific expert. Although this method shows improved performance even without further training compared to other model merging approaches, we show superior joint modeling of the sub-models by further fine-tuning the MoE architecture on a mixed dataset from various dialects.

**SLERP-based self-attention merging:** In our proposed methodology, the MLP layers of each expert within the MoE model are initialized using the MLP sub-layers from models previously trained on distinct dialects. For the self-attention sub-layers of the MoE model, we employ Spherical Linear Interpolation (SLERP) (Goddard et al., 2024; Shoemake, 1985) to merge the initial weights of the self-attention layers (Key, Value, and Query projections) across multiple dialects. SLERP allows for smooth, non-linear transitions between two weight vectors while preserving the intrinsic geometric properties of the spherical space. The process begins by normalizing the weights of the Key, Value, and Query layers from different dialect models to unit magnitude, ensuring that they lie on the surface of a unit sphere. Once normalized, the angle ($\theta$) between the weight vectors is computed using the dot product. If the vectors are nearly collinear (i.e., the dot product is close to 1), the merging process defaults to linear interpolation (LERP) for efficiency. Otherwise, SLERP calculates the scale factors based on the interpolation parameter $t$ and the angular separation between the vectors:

$$\text{SLERP}(t, \mathbf{v}_0, \mathbf{v}_1) = \frac{\sin((1-t)\theta)}{\sin(\theta)} \mathbf{v}_0 + \frac{\sin(t\theta)}{\sin(\theta)} \mathbf{v}_1$$

Where $v_0$ and $v_1$ represents the normalized weight vectors of the models. By merging the self-attention weights through SLERP, we can smoothly integrate the knowledge from different dialect-specific models into the initialization of the MoE model's self-attention layers, providing a more effective starting point for model training.

## 3 EXPERIMENTS

**Datasets:** We use benchmark datasets tailored for three dialects. For SQLite, we use two datasets from BIRD: 1) the development set and 2) the mini development set. For PostgreSQL, we utilize three benchmarks: 1) BIRD queries transpiled to PostgreSQL, 2) BIRD PostgreSQL mini development set, and 3) Pagila—a dataset specific to PostgreSQL containing real-world queries originally written for PostgreSQL, which are extracted from online resources. For BigQuery, we use two datasets: 1) BIRD queries transpiled to BigQuery, and 2) the GitHub_repositories dataset, a public BigQuery dataset featuring BigQuery-specific sample question/SQL pairs obtained from tutorials and online resources. Further details of the datasets are provided in Appendix A.2.

**Baselines:** In order to evaluate the quality of the synthetic queries generated with SQL-GEN, we compare the performance of the models trained on synthetic data considering the following datasets: 1) Gretel (Gretel, 2024): Gretel Text-to-SQL dataset consists >100K high-quality synthetic Text-to-SQL samples with a coverage across 100 distinct domains. 2) SQL create context (b mc2, 2023): This dataset consists ∼78K samples, obtained from the Spider (Yu et al., 2018b) and WikiSQL (Zhong et al., 2017) datasets by cleaning these sources. All queries were generated through a human-in-the-loop process, making it a strong baseline for comparing LLM-generated

data with human-annotated data. 3) BIRD train set (Li et al., 2024b): This dataset consists of ~10k human-annotated samples. Queries are considered more complex in comparison to the Spider and WikiSQL benchmarks.

For dialects other than SQLite, all queries from the baselines are transpiled to the target dialects to ensure their validity. Details of the models and metrics are provided in Appendix A.3 and details of the seed templates and tutorials are provided in Appendix A.4.

## 3.1 DIALECT-SPECIFIC SQL KEYWORDS CONVERGE IMPROVEMENT

We compare our generated SQL queries with two baseline datasets in terms of the diversity of queries, focusing on the use of unique SQL keywords and the frequency of dialect-specific queries. The results are presented in Figure 5. For an equitable comparison, we sample 60K queries from each baseline. Since our Text-to-SQL dataset exclusively contains SELECT queries, we exclude samples that do not start with the SELECT keyword. According to the results, our dataset exhibits the highest diversity and the greatest number of dialect-specific queries compared to the baselines. Interestingly, the SQL create context dataset, which is intended to be a SQLite dataset, contains several queries using the STRUCT() keyword, which is supported by BigQuery, not SQLite.

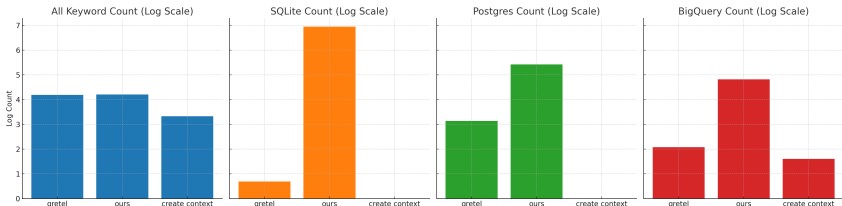

Figure 5: Comparison between queries generated by our method with the baselines in terms of diversity of the SQL keywords and number of dialect-specific queries in each of them.

## 3.2 POSTGRESQL RESULTS

For PostgreSQL, we train LoRA adapters for the CodeLlama 7B and Codestral 22B model on the transpiled baseline datasets and compared its performance against our proposed method. As shown in Table 1, our method achieves the highest performance on the PostgreSQL BIRD and Minidev benchmarks compared to other baselines, except for the BIRD train set. While training a model on the original BIRD training split delivers the highest performance on the BIRD development split, it significantly underperforms when evaluated on other PostgreSQL datasets, such as Pagila (as seen in the third row). This highlights the importance of diversity in training data to prevent overfitting to a specific distribution. In contrast, our approach achieves consistently high accuracy when evaluated on both the BIRD development split and other PostgreSQL datasets, demonstrating outstanding generalization ability. Moreover, these results demonstrate the importance of dialect specific datasets as the other transpiled queries couldn't match the performance of our method.

Table 1: Execution Accuracy (EX) of PostgreSQL Models on the three PostgreSQL benchmarks using CodeLlama 7B and Codestral 22B. "-" denotes the zero-shot performance of the models.

| Training Dataset | Benchmark | Model | EX (%) | ΔEX | Model | EX (%) | ΔEX |
|---|---|---|---|---|---|---|---|
| Bird train set | PostgreSQL BIRD | CodeLlama 7B | 44.37 | +20.08 | Codestral 22B | 52.26 | +5.68 |
| Our synthetic dataset | PostgreSQL BIRD | CodeLlama 7B | **39.22** | +14.93 | Codestral 22B | 49.84 | 3.26 |
| Gretel Text-to-SQL | PostgreSQL BIRD | CodeLlama 7B | 28.05 | +3.76 | Codestral 22B | 40.55 | -6.03 |
| SQL Create Context | PostgreSQL BIRD | CodeLlama 7B | 13.35 | -10.94 | Codestral 22B | 36.17 | -10.41 |
| - | PostgreSQL BIRD | CodeLlama 7B | 24.29 | 0 | Codestral 22B | 46.58 | 0 |
| Bird train set | PostgreSQL Minidev | CodeLlama 7B | 31.0 | +17.8 | Codestral 22B | 36.0 | +4.2 |
| Our synthetic dataset | PostgreSQL Minidev | CodeLlama 7B | **25.4** | +12.2 | Codestral 22B | **33.0** | +2.2 |
| Gretel Text-to-SQL | PostgreSQL Minidev | CodeLlama 7B | 14.6 | +1.4 | Codestral 22B | 23.0 | -8.8 |
| SQL Create Context | PostgreSQL Minidev | CodeLlama 7B | 7.8 | -5.4 | Codestral 22B | 25.2 | -6.6 |
| - | PostgreSQL Minidev | CodeLlama 7B | 13.2 | 0 | Codestral 22B | 31.8 | 0 |
| Bird train set | Pagila | CodeLlama 7B | 19.56 | -4.35 | Codestral 22B | 43.47 | -6.53 |
| Our synthetic dataset | Pagila | CodeLlama 7B | **39.13** | +15.22 | Codestral 22B | 50 | 0.0 |
| Gretel Text-to-SQL | Pagila | CodeLlama 7B | 36.95 | +13.04 | Codestral 22B | 50 | 0 |
| SQL Create Context | Pagila | CodeLlama 7B | 8.69 | -15.22 | Codestral 22B | 36.95 | -13.05 |
| - | Pagila | CodeLlama 7B | 23.91 | 0 | Codestral 22B | 50 | 0 |

## 3.3 BIGQUERY RESULTS

Similar to the PostgreSQL experiments, we present the results of the CodeLlama 7B and Codestral 22B model trained on various baseline datasets and evaluated on two BigQuery benchmark datasets: BIRD and the GitHub Repository database. Looking at the results provided in Table 2, consistent with the trends observed for the PostgreSQL dialect, on BigQuery BIRD, the model trained on our generated samples achieves the second highest performance, following the BIRD train set. For the GitHub Repository database, which is a BigQuery dialect specific dataset, our model outperforms the second-best model by a 10% margin, further demonstrating the effectiveness of our method to train dialect specific models.

Table 2: Execution Accuracy (EX) of BigQuery Models on the two BigQuery benchmarks using CodeLlama 7B and Codestral 22B. "-" denotes the zero-shot performance of the models.

| Training Dataset | Benchmark | Model | EX (%) | ΔEX | Model | EX (%) | ΔEX |
|---|---|---|---|---|---|---|---|
| Bird train set | BigQuery BIRD | CodeLlama 7B | 38.04 | +21.47 | Codestral 22B | 47.74 | +9.55 |
| Our synthetic dataset | BigQuery BIRD | CodeLlama 7B | **33.53** | +16.96 | Codestral 22B | **47.24** | +9.05 |
| Gretel Text-to-SQL | BigQuery BIRD | CodeLlama 7B | 26.73 | +11.16 | Codestral 22B | 36.74 | -1.45 |
| SQL Create Context | BigQuery BIRD | CodeLlama 7B | 10.84 | -5.73 | Codestral 22B | 39.19 | +1.0 |
| - | BigQuery BIRD | CodeLlama 7B | 16.57 | 0 | Codestral 22B | 38.19 | 0 |
| Bird train set | Github Repository | CodeLlama 7B | 7.5 | -7.5 | Codestral 22B | 7.5 | -12.5 |
| Our synthetic dataset | Github Repository | CodeLlama 7B | 25.0 | +10.0 | Codestral 22B | **30.0** | +10.0 |
| Gretel Text-to-SQL | Github Repository | CodeLlama 7B | 17.5 | +2.5 | Codestral 22B | 22.5 | +2.25 |
| SQL Create Context | Github Repository | CodeLlama 7B | 0.0 | -15.00 | Codestral 22B | 20.0 | 0 |
| - | Github Repository | CodeLlama 7B | 15 | 0 | Codestral 22B | 20.0 | 0 |

## 3.4 SQLITE RESULTS

Utilizing SQL-GEN, we generate 20K samples for the SQLite dialect. Appendix A.4.2 studies the impact of the number of samples. We train three different models with different sizes from 7B to 22B on these samples. For a fair comparison with the baselines, we only use the Spider databases for generating the synthetic data. For this comparison, we train models on: 1) The entire BIRD training set; 2) 20K samples from the SQL Create Context (b mc2, 2023); and 3) 20K samples from the Gretel Text-to-SQL datasets (Gretel, 2024). We assess the Text-to-SQL performance of these models on the BIRD development set and minidev set (see Table 3). Additionally, we evaluate the zero-shot performance of each model and calculate the performance gains for each method relative to zero-shot.

SQL-GEN generated samples significantly surpass the Gretel dataset, achieving a large performance gain of approximately 10% across all model sizes. Furthermore, LLMs trained on SQL-GEN synthetic data consistently outperform those trained on the human-annotated SQL Create Context data, underscoring the high quality of SQL-GEN synthetic data. While LLMs trained on the BIRD dataset consistently exhibit the highest performance on BIRD development sets, this outcome is likely due to overfitting to the canonical input distribution of the BIRD train set which is similar to its development set (Yu et al., 2020).

Table 3: Execution Accuracy (EX) of SQLite Models on the BIRD development set and minidev set using CodeLlama 7B, CodeGemma 7B, and Codestral 22B Models. "-" denotes the zero-shot performance of the models.

| Training Dataset | Model | Dataset | EX (%) | ΔEX | Dataset | EX (%) | ΔEX |
|---|---|---|---|---|---|---|---|
| Bird train set | CodeLlama 7B | dev set | 40.22 | +22.36 | minidev set | 38.4 | +24.8 |
| Our synthetic dataset | CodeLlama 7B | dev set | **38.33** | +20.47 | minidev set | **30.00** | +16.4 |
| Gretel Text-to-SQL | CodeLlama 7B | dev set | 26.01 | +8.15 | minidev set | 19.6 | +6.0 |
| SQL Create Context | CodeLlama 7B | dev set | 18.31 | +0.45 | minidev set | 12.6 | -1.0 |
| - | CodeLlama 7B | dev set | 17.86 | 0 | minidev set | 13.6 | 0.0 |
| Bird train set | CodeGemma 7B | dev set | 45.63 | +11.87 | minidev set | 40.4 | +10.4 |
| Our synthetic dataset | CodeGemma 7B | dev set | **42.37** | +8.64 | minidev set | **36.4** | +6.4 |
| Gretel Text-to-SQL | CodeGemma 7B | dev set | 30.83 | -2.93 | minidev set | 30.6 | +0.6 |
| SQL Create Context | CodeGemma 7B | dev set | 28.87 | -4.89 | minidev set | 29.6 | -0.4 |
| - | CodeGemma 7B | dev set | 33.76 | 0 | minidev set | 30.0 | 0.0 |
| Bird train set | Codestral 22B | dev set | 53.12 | +8.6 | minidev set | 50.4 | +10.0 |
| Our synthetic dataset | Codestral 22B | dev set | **50.45** | +5.93 | minidev set | **46.6** | +6.2 |
| Gretel Text-to-SQL | Codestral 22B | dev set | 37.87 | -6.65 | minidev set | 30.8 | -9.6 |
| SQL Create Context | Codestral 22B | dev set | 40.80 | -3.72 | minidev set | 36.8 | -3.6 |
| - | Codestral 22B | dev set | 44.52 | 0 | minidev set | 40.4 | 0.0 |

To pinpoint whether the gains are consistent across, we evaluate different models on different SQL query complexity levels: simple, medium, or challenging, as presented in Appendix A.5.

**Database Adaptation:** SQL-GEN operates independently of specific databases, enabling the generation of high-quality synthetic data for any database. Therefore, as another use case of synthetic data, we introduce *Database Adaptation*, to improve the performance in cross-domain Text-to-SQL setting. This involves generating synthetic queries for databases for which no pre-existing question-SQL pairs are available. We apply this training in two distinct ways: (1) in-context learning, which leverages the generated queries directly within the model's input context as demonstrations, and (2) model tuning, which involves supervised fine-tuning of the model weights:

**Database Adaptation With Model Tuning:** Our synthetic data generation pipeline is designed to generate question-SQL pairs for any database. To demonstrate this, we generated 10K pure synthetic question-SQL pairs across the 11 databases in the BIRD development set and separately 10K samples for the entire databases in the BIRD training set using Gemini-1.5-pro. We then compared the performance of two model against a model trained on the original 10K training samples from the BIRD benchmark. The results are detailed in Section 3.4. The table indicates that our synthetic generation approach on the BIRD development set databases achieves performance comparable to the original BIRD training set with only 1.5% gap. This is particularly noteworthy given that generating synthetic samples is significantly less resource- and cost-intensive compared to creating 10,000 human-annotated samples. The latter involves 11 crowd-source workers to annotate the samples. Additionally, synthetic data generation on development split outperforms train split showing that SQL-Gen helps to learn unseen database and improve performance.

| Training Data | EX (%) | ΔEX |
|---|---|---|
| BIRD train set | 40.22 | +22.36 |
| Synthetic sample on BIRD dev dbs | 38.78 | +20.92 |
| Synthetic sample on BIRD train dbs | 34.68 | +16.82 |
| Zero-shot (no training) | 17.86 | 0 |

Table 4: Using our proposed pipeline to generate 10K synthetic data for BIRD development set databases.

| #ICL | Model | EX (%) | ΔEX |
|---|---|---|---|
| Zero-shot | CodeLlama | 12.35 | 0.0 |
| 1 | CodeLlama | 17.97 | +5.62 |
| 5 | CodeLlama | 20.22 | +7.87 |
| 10 | CodeLlama | 22.47 | +10.12 |

Table 5: Using SQL-GEN to generate synthetic data for the given database. #ICL denotes the number of in-context learning samples used in the prompts.

**Database Adaptation with In-context Learning:** An alternative method to enhance the performance of LLMs on task-specific datasets is through in-context learning (Brown et al., 2020). We explore the concept of database adaptation through in-context learning, using synthetic queries as few-shot in-context samples without additional model training. To evaluate this approach, we generate 500 synthetic samples for the California schools database from the BIRD development set. We then test the model's performance on 89 samples from this database using different numbers of in-context samples. The results, presented in Section 3.4. For selecting the few-shot samples, we use cosine similarity between question embeddings. These results demonstrate that we can achieve a 10% improvement in accuracy without any training.

**Data Augmentation:** Beyond merely creating a pool of pure synthetic question-SQL pairs for training, synthetic data generation offers the potential to augment existing datasets (e.g. mixing with original dataset), thereby enhancing model performance beyond what is achievable with solely the available data. We consider integrating synthetic data generated for specific target databases (as discussed in Database

Table 6: Performance comparison of the data augmentation method on the BIRD development set using different LLMs.

| Training Dataset | Model | EX (%) | ΔEX |
|---|---|---|---|
| BIRD train set | CodeLlama 7B | 40.22 | 0 |
| BIRD Train + synthetic | CodeLlama 7B | 45.82 | +5.6 |
| BIRD train set | CodeGemma 7B | 45.63 | 0 |
| BIRD Train + synthetic | CodeGemma 7B | 51.10 | +5.47 |
| BIRD train set | Codestral 22B | 53.12 | 0 |
| BIRD Train + synthetic | Codestral 22B | 56.45 | +3.33 |

Adaptation, see Section 3.4) with pre-existing training datasets. To this end, we merge 10K synthetic question-SQL pairs generated on the BIRD development databases with the 10K pairs from the BIRD training set. We then train various models using this combined dataset and compared their performance to models trained solely on the original BIRD training set. For a balanced comparison, models using the combined datasets are trained for only one epoch, whereas those trained exclusively on the BIRD training set are trained fro two epochs. As shown in Table 6, augmenting

training data results in a performance improvement of up to 5.6%, a significant enhancement compared to previous work, such as Yang et al. (2024) (which demonstrated only a 1.5% improvement in performance after augmentation on the same base model CodeLLama 7B).

## 3.5 EXPERTS MERGING RESULTS

We evaluate various expert merging approaches for integrating dialect-specific models into a single unified model and compare to our method based on the Mixture of Experts (MoE) architecture. We utilize three expert CodeLlama 7B models, each trained on synthetic question-SQL pairs for SQLite, PostgreSQL, and BigQuery. We consider three popular model merging techniques: DARE (Yu et al., 2024), TIES (Yadav et al., 2024), and SLERP. Unlike the first two, SLERP can only merge two models at a time. Therefore, we initially merge the SQLite and PostgreSQL experts and then combined the resulting model with the BigQuery expert. Additionally, we fine-tune a *generalist* (not dialect-specific) CodeLlama 7B and MoE 3x7B model on 40K samples from a mix of different dialects to establish a baseline for comparison. The *generalist MoE* baseline is a MoE 3x7B model initialized from CodeLlama 7B model and trained with 40K combined dialect samples for 1 epoch. We compare all these methods to the proposed method for initializing the MoE model which is trained only for one epoch of 20K samples from different dialects. We train for a single epoch to promote effective collaboration among the submodules. We also include the performance of the proposed MoE model before the single epoch fine-tuning to better understand understand the effectiveness of our proposed initialization method. We assess the performance of the different models on PostgreSQL's Pagila, BigQuery's Github Repository, and 10% of random samples from the SQLite BIRD dev set, with results detailed in Table 7. The table demonstrates that the proposed MoE model overall outperforms others, even exceeding the performance of the individual dialect experts, highlighting the effectiveness of our approach in sharing common SQL knowledge across dialects while preserving dialect-specific expertise. The MoE architecture also enhances the model's learning capacity, contributing to improved overall performance. Notably, our initialization method is more effective at maintaining high dialect-specific performance compared to the generalist MoE 3x7B model. Among all merging techniques, SLERP achieves the highest performance, surpassing even the generalist model trained on the combined dialect-specific datasets, which is the main reason for initializing the self-attention sub-layers. Moreover, the results suggest that our proposed method for initialization even before fine-tuning provide a strong baseline surpassing TIES and DARE methods for model merging. In Appendix A.8, we provide detailed analysis of token-level routing for MoE architecture.

Table 7: Comparison between different dialect expert merging approaches and our proposed MoE for dialect benchmarks. Generalist model refers to CodeLlama trained on the combination of the dialect datasets.

| Model | BIRD SQLite | PostgreSQL Pagila | BigQuery BIRD | Overall |
|---|---|---|---|---|
| CodeLlama 7B SQLite expert | 34.01 | 39.13 | 25 | 32.71 |
| CodeLlama 7B Postgres expert | 29.93 | 39.13 | 20 | 29.68 |
| CodeLlama 7B BigQuery expert | 33.33 | 32.60 | 27.5 | 31.14 |
| CodeLlama 7B generalist | 33.33 | 32.66 | 32.25 | 32.83 |
| Merged experts + SLERP | 34.69 | 39.13 | 27.5 | 33.77 |
| Merged experts + TIES | 35.37 | 30.43 | 27.5 | 31.1 |
| Merged experts + DARE | 35.37 | 36.95 | 17.5 | 29.94 |
| MoE 3x7B (ours) | 36.05 | 39.13 | 22.5 | 32.56 |
| MoE 3x7B fine-tuned (ours) | 34.69 | 39.13 | 32.25 | **35.44** |
| MoE 3x7B generalist | 34.01 | 41.3 | 25 | 33.43 |

## 4 CONCLUSIONS

We present a novel framework for generating dialect-specific synthetic data to tackle the diverse SQL dialect modeling challenges for Text-to-SQL. The proposed framework addresses the unique challenges such as keywords and functions being different for each SQL dialect, constituting a scalable approach. It significantly narrows the performance gap with human-annotated datasets and creates the highest quality datasets for other dialects. Our comprehensive evaluations across three models and multiple benchmarks, showcase the effectiveness of the proposed data generation framework. Additionally, our innovative approach integrates dialect-specific experts into a unified model, enhancing performance by promoting effective information sharing among them.

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

# A APPENDIX

## A.1 RELATED WORK

### A.1.1 SYNTHETIC DATA GENERATION

Early work for data augmentation for Text-to-SQL largely rely on human annotations to verify the generated SQL queries or extract high-quality question-SQL pairs (Iyer et al., 2017; Yu et al., 2018a). Guo et al. (2018) use a pattern-based approach to generate SQL queries and utilize a copy-based Seq2Seq model to directly translate SQL queries into natural language questions. Some of the recent methods (Wu et al., 2021; Yu et al., 2020; Zhao et al., 2022; Wang et al., 2021) rely on grammar-based approaches to generate question-SQL pairs. Wu et al. (2021) use an abstract syntax tree grammar to generate SQL queries and then employs a hierarchical SQL-to-question generation model to obtain questions for the SQL queries. Similarly, Yu et al. (2020) extract and manually annotate question and SQL templates from Spider (Yu et al., 2018b) to induce a grammar, then use the grammar to generate synthetic samples for databases in Spider and WikiTables (Bhagavatula et al., 2015). However, all methods relying on grammars have the drawback of generating samples that lack diversity and highly depend on the grammar used (Yu et al., 2020), which makes them not suitable for tasks that require generalization to new schemas.

Recently, Li et al. (2024a) propose a bidirectional method with question-to-SQL and SQL-to-question augmentation. In the former, they use some human-annotated samples with in-context learning with LLMs to generate queries for a new database, and in the latter, they extract templates from Spider and fill those templates with the schema of a given database. This method has the limitation that the diversity of the question and SQL pairs is restricted to either templates or in-context samples. Concurrently with our work, SENSE (Yang et al., 2024) proposed a two-step synthetic data generation process to enhance the performance of open-source text-to-SQL models. In the first step, they utilize a robust LLM to generate a supervised fine-tuning dataset with a single LLM call. In the second stage, they employ a smaller, weaker LLM to produce some incorrect SQL queries, which are then used to construct a preference dataset. The initial phase of their method is similar to our proposed approach; however, their method's simplicity, which lacks execution result filtering or conditioning on externally provided SQL keywords and relies solely on the LLMs' parametric knowledge, contrasts with our method that incorporates external knowledge to craft diverse queries. Lastly, Gretel (2024) release a high-quality large dataset of 100K question-SQL pairs from different domains.[1] Overall, none of the previously mentioned approaches consider different dialects and they are proposed for SQLite [2], which is a significant drawback of their work.

In the domain of synthetic data generation for code, recent work such as Reflexion (Shinn et al., 2023) leverage external or internal feedback signals to enhance the code reasoning capabilities of language models. Code Alpaca features a dataset of 20K code instructions automatically generated by applying SELF-INSTRUCT (Wang et al., 2022) to LLMs across different seed tasks. Wizard-Coder (Luo et al., 2023) introduces Code Evol-Instruct, which uses manually crafted prompts to guide LLMs, thereby increasing the complexity and diversity of the synthetic data. Similarly, Magicoder (Wei et al., 2023) proposes OSS-INSTRUCT, which consists 75K diverse synthetic instruction samples from open-source code snippets that are used as the seeds to both increase diversity and also control the data generation process.

### A.1.2 MODEL MERGING

Training specialized, task-specific models presents several challenges, including the storage costs associated with maintaining multiple models, the substantial memory requirements for deploying these models, and the rapid obsolescence of models as training datasets age. One proposed solution to mitigate these issues is model merging (Goddard et al., 2024). Initial approaches to model merging, such as Task Arithmetic (Ilharco et al., 2022), involve calculating task-specific vectors by determining the weight differences between the fine-tuned model and its base counterpart. These vectors are then linearly combined and reintegrated with the original base model. Subsequent methodologies like DARE, TIES, and Model BreadCrumbs (Yadav et al., 2024; Yu et al.,

---

[1]The methodology to generate the pairs is not publicly available.

[2]Gretel dataset doesn't specify the dialect.

2024; Davari & Belilovsky, 2023) have aimed to minimize interference among task-specific models through techniques such as sparsification, sign consensus algorithms, and the exclusion of extreme values. Additionally, DARE introduces random pruning to align more closely with the base model's performance (Goddard et al., 2024). More recently, the integration of model merging with Mixture of Experts (MoE) architectures has been explored. This method, termed FrankenMoEs, initializes MoE MLP layers using weights from task-specific models (Goddard, 2024; Tang et al., 2024). Our work extends these efforts by specifically leveraging features from dialect-specific models for gate initialization and merging self-attention sublayers within transformer architectures.

## A.2 Datasets Details

### A.2.1 SQLite

To the best of our knowledge, the majority of large-scale, cross-domain Text-to-SQL datasets are tailored for the SQLite dialect. Among these, the Spider (Yu et al., 2018b) and BIRD Li et al. (2024b) datasets are two popular benchmarks used to evaluate Text-to-SQL model performance (Pourreza & Rafiei, 2024; Wang et al., 2023; Talaei et al., 2024; Li et al., 2024a), establishing them as primary standards in this area. We use the Spider training set to derive seed templates. To ensure a fair comparison, we report the results using the BIRD benchmark for the SQLite dialect, with the Spider dataset serving as a baseline to assess the quality of our synthetic samples. The BIRD benchmark includes two development sets: the original dev set, which contains 1534 question-SQL pairs with some incorrect SQL queries Li et al. (2024a), and the minidev set, which features smaller size of 500 higher quality question-SQL pairs. We evaluate on both.

### A.2.2 PostgreSQL

As mentioned in the previous section, there is a shortage of human-annotated benchmarks for dialects other than SQLite. Therefore, for PostgreSQL dialect, we use the following datasets to compare the performance of the models:

**PostgreSQL BIRD:** All 11 databases in the BIRD development set are migrated from SQLite to PostgreSQL, and their SQL queries are transpiled to PostgreSQL using Mao (2023). This migration and transpilation are conducted under a best-effort setting. However, some challenges are encountered: a few databases have foreign key violations, and some queries cannot be successfully transpiled to PostgreSQL. Out of the 1534 samples in the development set, 951 queries are successfully migrated for PostgreSQL.

**PostgreSQL MiniDev:** Similar to the approach we use for the PostgreSQL BIRD dataset, the authors of BIRD transpile queries in the minidev set, manually annotating any pairs that cannot be directly translated from SQLite to PostgreSQL. This dataset comprises 500 question-SQL pairs.

**Pagila:** Since the BIRD benchmark was originally developed for SQLite, the transpiled queries do not utilize many PostgreSQL-specific functions and keywords. To address this, we created a PostgreSQL-specific benchmark, Pagila (Gunduz). The Pagila database mimics a real-world business by modeling a DVD rental store. It includes tables for films, actors, customers, inventory, rental transactions, and more, making it a useful resource for educational purposes. This database is designed to provide a standard schema for use in books, tutorials, and articles. We gathered a dataset of 46 human-annotated question-SQL pairs, which were validated and extracted from open-source resources for this database.

### A.2.3 BigQuery

We use the following baselines for reporting the performance for BigQuery dialect:

**BigQuery BIRD:** Similar to the approach mentioned for PostgreSQL, all 11 databases in the BIRD development set are migrated from SQLite to BigQuery, and their SQL queries are transpiled to BigQuery using Mao (2023). Out of the 1534 samples in the development set, 1309 queries are successfully migrated for BigQuery.

**Github Repositories:** In our work, for the BigQuery-specific database, we utilized one of the publicly available and widely used databases, the GitHub repositories (Cloud). This database allows for monitoring and analyzing GitHub's activity since 2011. We gathered a dataset of 40 human-annotated question-SQL pairs, validated and extracted from open-source resources for this database.

### A.3 MODELS & METRICS

#### A.3.1 MODELS

To evaluate the quality of the generated samples, we fine-tune models from different families, including CodeLlama 7B (Roziere et al., 2023), CodeGemma 7B (cod), and Codestral 22B (Mistral, 2024), using LoRA adapters for all linear layers (Hu et al., 2021) with a rank of 128 and alpha of 256. For the synthetic data generation process we use Gemini 1.5 pro as the main model and Gemini 1.5 flash as the quality check model. To ensure the data generation process is affordable and replicable, we also include high-performing, open-source LLMs for synthetic data generation. For template expansion and filling, we employ Llama-3-70B (met), and for quality check step, we employ Mixtral-8x7B (Jiang et al., 2024).

#### A.3.2 METRICS

We primarily focus on execution accuracy (EX) as the main metric, which is widely accepted as the standard for all Text-to-SQL benchmarks (Yu et al., 2018b; Li et al., 2024b).

### A.4 METHOD SEEDS

In this section, we present details regarding the number of seed SQL templates extracted from the Spider train set, which comprises 8,659 training examples across 146 databases. To generate seed templates for dialects other than SQLite, we transpiled the queries from SQLite to the target dialects using SQLGlot. Table 8 provides the counts of seed SQL queries for each dialect. Moreover, for scraping the tutorials we used the following websites for each dialect:

- SQLite: SQLite tutorial
- PostgreSQL: PostgreSQL tutorial
- BigQuery: BigQuery syntax

Table 8: Number of seed SQL templates extracted from the Spider training dataset for three dialects of SQLite, BigQuery, PostgreSQL.

| Dialect | Number of templates |
| --- | --- |
| SQLite | 1458 |
| BigQuery | 1665 |
| PostgreSQL | 1293 |

### A.4.1 QUALITY CHECK ABLATION

In our proposed method, we opted to use a secondary LLM to act as a judge in the quality check step, ensuring the high quality of the generated samples and avoiding repetition of previous errors. In this section, we assess this approach by comparing two scenarios: one where the same LLM acts as judge, and another where a secondary LLM performs the judging role. The results, presented in Table 9, demonstrate that the CodeLlama 7B model trained on the dataset filtered by a secondary model achieved higher performance on the BIRD development set, thus validating our strategy. Moreover, Table 10 provides the result of removing the quality check step and shows a performance drop in accuracy, validating the importance of this step to remove low quality samples.

Table 9: Performance comparison between two scenarios, when the same model generates and filters candidate samples, and another when a secondary model is used for filtering.

| Base Model | Judge Model | EX (%) |
|---|---|---|
| Mixtral 8x7B | Mixtral 8x7B | 32.59 |
| Llama 3 70B | Llama 3 70B | 33.41 |
| Llama 3 70B | Mixtral 8x7B | **34.55** |

Table 10: Performance on the ablation of the quality checker model with Codellama 7B on BIRD dev set. OS refers to using open-source models like Llama3 and Mixtral for data generation.

| Pipeline | EX (%) |
|---|---|
| Pipeline without quality check (OS) | 32.85 |
| Full pipeline (OS) | 34.55 |

### A.4.2 THE IMPACT OF THE SAMPLE SIZE

Due to the limited availability of large-scale benchmarks for dialects other than SQLite, our ablation studies focus solely on the SQLite dialect. For each target dialect, we use our method to generate 20K samples. We assess the impact of varying sample sizes on the final performance of the model. Table 11 presents the performance with the CodeLlama 7B model when trained on different sample sizes generated by Llama 3 and Mixtral models, and tested on the BIRD development set. As indicated, there is diminishing return in performance as the sample size increases.

Table 11: Evaluating the performance of CodeLlama 7B using different sample sizes on BIRD development set. "-" denotes the zero-shot performance of the model

| #Samples | Model | EX (%) | $\Delta$EX |
|---|---|---|---|
| - | CodeLlama | 17.86 | 0 |
| 5000 | CodeLlama | 32.59 | + 14.73 |
| 10000 | CodeLlama | 33.57 | +15.71 |
| 20000 | CodeLlama | 34.55 | +16.69 |

### A.5 COMPLEXITY ANALYSIS

For complexity analysis we used official BIRD classification based on the number and type of the SQL keywords used in the ground truth SQL query for each question in BIRD development set. The results are provided in the Table 12 for the two synthetic and human annotated baselines together with the zero-shot performance of CodeLlama 7B model. Based on the results model trained on our synthetic data has the highest performance across all of the complexity levels. Interstingly, due to the simplisity of the samples in the SQL create context dataset performance on the challenging samples is even lower than the zero-shot baseline.

Table 12: Comparison of different datasets across varying SQL query complexities on the BIRD development set for CodeLlama 7B trained on each dataset. "-" denotes the zero-shot performance of the models

| Training Dataset | Model | simple EX (%) | moderate EX (%) | challenging EX (%) |
|---|---|---|---|---|
| Our synthetic dataset (Gemini) | CodeLlama | **49.51** | **24.08** | **12.5** |
| Gretel Text-to-SQL | CodeLlama | 31.78 | 11.61 | 11.8 |
| SQL Create Context | CodeLlama | 25.18 | 9.67 | 2.08 |
| - | CodeLlama | 24.1 | 7.95 | 9.72 |

### A.6    SAMPLE GENERATION FILTERS

#### A.6.1    EXECUTION CHECK

Unlike the template expansion step, SQL queries in this stage are generated from actual databases, allowing us to execute the queries over the databases. This capability enables us to utilize dialect-specific database engines to discard samples that are syntactically incorrect. This method is more robust than the parsing checks with SQLglot, as employed in Gretel (2024), providing a more effective way to ensure the accuracy of our SQL queries.

#### A.6.2    QUESTION-SQL MISMATCH

During the query generation process using various LLMs such as Gemini (Team et al., 2023), GPT-3.5 Turbo, and Llama-3-70B (met), we observe a recurring issue where some mismatches occurred between the conditions in the generated SQL queries and the corresponding questions. To minimize these mismatches, we develop a set of validator functions to detect inconsistencies. For each generated SQL query, we extract all conditions that correspond to database values using a SQL parser. We then calculate the maximum semantic similarity and the minimum edit distance between these conditions and all keywords in the question. SQL queries where a keyword's minimum distance exceeds a threshold $\beta_1$ or whose maximum semantic similarity is below another threshold $\beta_2$ are discarded. Figure 6 illustrates an example of question-SQL pair which is rejected because of mismatch between questions and SQL outputs.

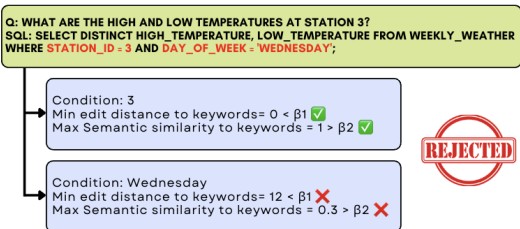

Figure 6: An example of a filtered question-SQL pair due to question and SQL mismatch.

#### A.6.3    AGGREGATION CHECK

Another consistent issue with the LLMs was the inappropriate use of aggregation functions on columns that already contain aggregated values. For example, in response to the question, "What are the average ages of singers?" the LLM might generate: "SELECT AVG(average_age) FROM singer", where there is a redundant aggregation function. To address these cases, we examine the SQL queries for aggregation functions. If the column name already includes an aggregation function in its name, we discard those queries.

#### A.6.4    DEDUPLICATION AND LENGTH CHECK

Similar to the approaches proposed in Wei et al. (2023); Wang et al. (2022), we discard duplicated SQL queries and pairs where the question length exceeds a specific threshold, $\alpha_1$.

## A.7 DIALECT SPECIFIC KEYWORDS

This section presents some examples of dialect-specific keywords for BigQuery, PostgreSQL, and SQLite. These keywords, listed in Table 13, are not supported interchangeably among the three dialects. These keywords are just samples of dialect specific keywords and there are many more dialect specific keywords and functions.

Table 13: List of some of SQL keywords that are not supported entirely across all three dialects of BigQuery, PostgreSQL, and SQLite.

| Keyword | SQLite | PostgreSQL | BigQuery |
|---|---|---|---|
| *CREATE MODEL* | × | × | ✓ |
| *ML.TRANSLATE* | × | × | ✓ |
| *ML.GENERATE_TEXT* | × | × | ✓ |
| *ML.ANNOTATE_IMAGE* | × | × | ✓ |
| *SAFE* | × | × | ✓ |
| *QUALIFY* | × | × | ✓ |
| *WITH OFFSET* | × | × | ✓ |
| *ARRAY_AGG* | × | ✓ | ✓ |
| *STRUCT* | × | × | ✓ |
| *ILIKE* | × | ✓ | × |
| *LATERAL* | × | ✓ | × |
| *SERIAL* | × | ✓ | × |
| *CTID* | × | ✓ | × |
| *PRAGMA* | ✓ | × | × |
| *REGEXP_CONTAINS* | × | × | ✓ |
| *REGEXP_MATCHES* | × | ✓ | × |
| *GLOB* | ✓ | × | × |
| *JULIANDAY* | ✓ | × | × |
| *DATE_TRUNC* | × | ✓ | × |
| *TIMESTAMP_TRUNC* | × | × | ✓ |

## A.8  MoE Analysis

We analyze the hidden representations of our proposed MoE model and compare it with the baseline generalist MoE model across three distinct layers: Layer 1, Layer 16, and Layer 32. Although both MoE models are trained with a load balancing loss, our initialization approach for the gates leads to an expert collapse in the middle layers. This issue primarily stems from the high similarity in the hidden representations of the positive prompts for each dialect expert across all layers. Additionally, similar to the experiments conducted with the original Mixtral model (Jiang et al., 2024), there is no distinct expert associated with different token types in either our MoE model or the baseline generalist MoE model.

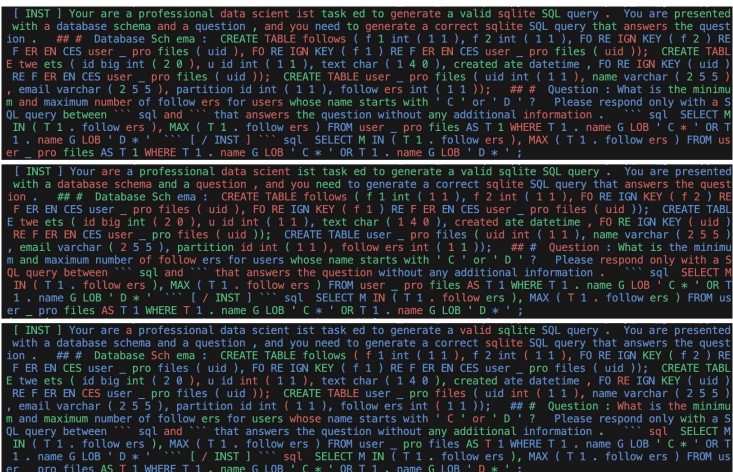

Figure 7: Token routing for the MoE model, initialized from CodeLlama and trained on a 40K samples dataset. The top figure illustrates Layer 1, the middle figure shows Layer 16, and the bottom figure corresponds to Layer 32.

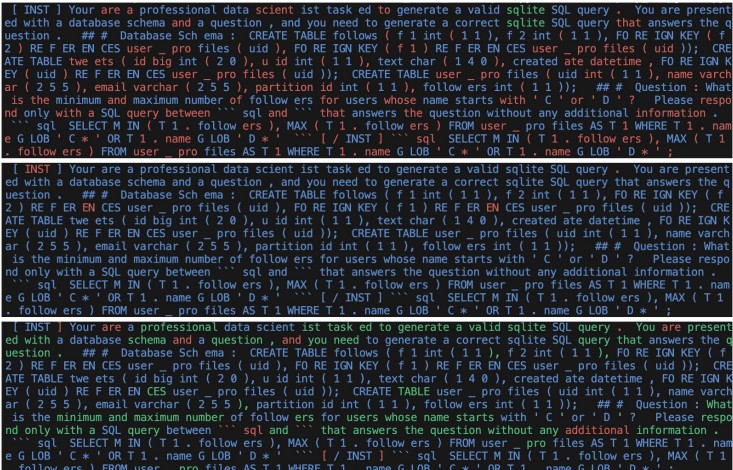

Figure 8: Token routing for our proposed method for initializing the MoE model, and trained on a 20K samples dataset. The top figure illustrates Layer 1, the middle figure shows Layer 16, and the bottom figure corresponds to Layer 32.

## A.9 PROMPT TEMPLATES

This section provides the detailed prompts used in this work for each of the step in our work.

### A.9.1 TEMPLATE EXPANSION

This section provides the prompt for template expansion step, Figure 9, where a seed template together with a sampled dialect-specific tutorial is passed to the LLM and asked to generate a new dialect-specific template. Additionally Figure 3 provides an example of template expansion for BigQuery dialect.

```
 You are an agent expert in data science and SQL.

Your are tasked with increasing the complexity of a given SQL
query template by inspiring from a sample tutorial document for
{DIALECT} SQL.

A query template is defined as a SQL query with placeholders
for columns, tables, and literals.  For each template, you will
be provided with:
1.  SQL keywords and functions.
2.  column or alias.column which is a placeholder for a column
name.
3.  table which is a placeholder for a table name.
4.  literal which is a placeholder for a literal value that can
be a string, number, or date.

Next you will be provided with a tutorial doc for {DIALECT} SQL
and a SQL query template.  You have to increase the complexity
of the query template by adding more SQL keywords and functions
inspired from the tutorial doc.

Tutorial:
{TUTORIAL}

Query template:
{QUERY_TEMPLATE}

Your response should be a valid {DIALECT} SQL template with
column, table, and literal placeholders.  Do not fill the
placeholders.

Your response should be only a valid JSON object as follows
without any additional text:
{{
reasoning:  Your step by step reasoning for increasing the
complexity of the query template by using the tutorial doc.,
query_template:  A valid SQL query template with placeholders
}}
```

Figure 9: Prompt used for the template expansion step (PTempGen)

A.9.2 SAMPLE GENERATION

In this section, we provide the prompt for the sample generation step, Figure 10, where a dialect-specific template together with a database schema are passed to the LLM and asked to generate question/SQL pair. Additionally Figure 11 provides an example of sample generation step.

```
 You are an agent expert in data science and SQL.

You are provided with a database schema together with a
{DIALECT} SQL template with placeholders.
Your job is to create synthetic data for training a Text-to-SQL
model.
Having the database schema and the SQL template, you should get
inspired by the SQL template to generate a business question
that a user might ask from the given database.

Always make sure that the SQL query is in the correct syntax
and it extracts meaningful and logical information for analysis.
The final SQL query should be a valid {DIALECT} SQL query
without any placeholders.
Make any necessary changes to the SQL template to fit the
database schema.  The SQL query should be able to answer the
business question.
You will be penalized for useless or meaningless queries.
The question should be generated as if it is asked by a user
who do not know the database schema and it should be clear and
concise.
You don't have to use all of the keywords in the SQL template,
but you should use at least some of them that are relevant to
the business question.
Make sure all of the conditions are correct, specificallty when
you are using operators, make sure types are compatible.
All of the conditions in the SQL query should be explicitly
mention in the question and avoid unnecessary conditions.
Question shouldn't be too simple or too complex.  It should be
meaningful and exact without any ambiguous terms.

Datbase schema:
{DATABASE_SCHEMA}

{DIALECT} SQL template to get inspired by:
{SQL_TEMPLATE}

Thank step by step about how to effectively generate a
meaningful business question from the SQL template and the
database schema.
{{
question:  The bussiness question that a user might from the
given database and the answer expects a SQL query similar to the
SQL template provided.,
sql_query:  A valid {DIALECT} SQL query that answers the business
question.
}}
```

Figure 10: Prompt used for the sample generation (PGen)

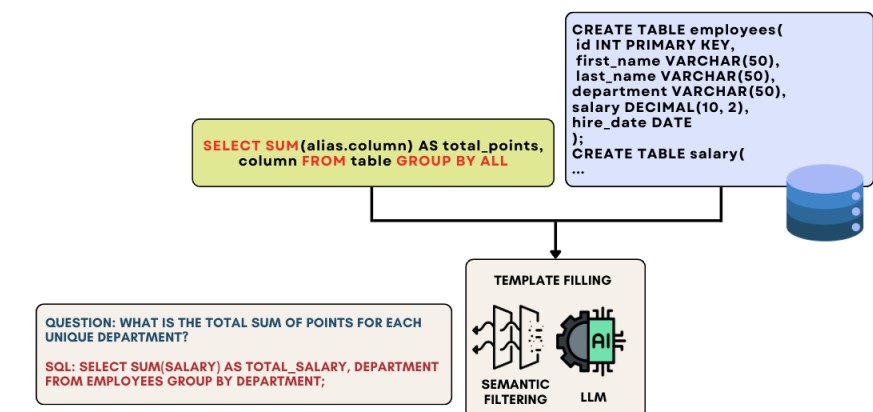

Figure 11: An example of sample generation using a random database and sampled SQL template.

A.9.3  QUALITY CHECK

This section outlines the template for the quality check prompt, Figure 12. The template receives a database schema, a generated question, a generated SQL query, and the execution result of the query. It then identifies and resolves any semantic discrepancies between the pair.

```
You are a meticulous data quality assurance professional.

Your job is to ensure that a dataset has high quality since it
is going to be used for training models.
You are presented with a database schema and a {DIALECT} SQL
query, its results, and a question.
If the pair needs fixing, you should fix the question or SQL
query to make it acceptable.

You have to make sure the following items are satisfied:
1.  Question should match the {DIALECT} SQL query, and it
shouldn't be ambiguous.  The question should be asked as if a
non-technical person without access to the database is asking
it.
2.  The SQL query should exactly answer what is mentioned in the
question, without any additional or irrelevant information.

If question is not answerable or is ambiguous, change the
question based on the database schema, then answer the new
question with a new SQL query.
If SQL query is not correct, fix the SQL query based on the
database schema, then answer the question with the fixed SQL
query.

DATABASE_SCHEMA:
{DATABASE_SCHEMA}

Question:
{QUESTION}

{DIALECT} SQL Query:
{SQL_QUERY}

{DIALECT} SQL Query Result:
{SQL_QUERY_RESULT}
Your response should be only a valid JSON object as follows
without any additional text:
{{ reasoning:  Your step by step reasoning for deciding if the
question or SQL query needs fixing.
fixing_needed:  YES or NO,
fixed_question:  If the question is not acceptable, provide a
fixed version of the question.,
fixed_sql_query:  If the SQL query is not acceptable, provide a
fixed version of the SQL query.
}}
```

Figure 12: Prompt used for the Quality Check step (PQuality)

