# OpenReview forum: "SQL-GEN: Bridging the Dialect Gap for Text-to-SQL Via Synthetic Data And Model Merging"
_ICLR.cc/2025/Conference — Submitted to ICLR 2025_

### Official Review · Reviewer_a2FH · 2024-11-01

**Soundness:** 3
**Presentation:** 3
**Contribution:** 3
**Rating:** 6
**Confidence:** 5

**Summary:**

The paper addresses the challenges of adapting Text-to-SQL systems across SQL dialects like BigQuery, PostgreSQL, and SQLite. Traditional Text-to-SQL models, optimized for SQLite, struggle with dialect-specific syntax variations, making translation and model adaptability challenging. SQL-GEN proposes a synthetic data generation method, SQL-GEN, which leverages dialect-specific SQL tutorials to create diverse, high-quality training samples for any SQL dialect. It introduces a Mixture of Experts (MoE) approach for merging dialect-specific models, enhancing multi-dialect capability.

**Strengths:**

1. SQL-GEN’s synthetic data generation effectively addresses the lack of annotated data for various SQL dialects, enhancing Text-to-SQL systems' adaptability across dialects without needing human annotations. And it is the first work to discuss its important problem.
2. The moe architecture efficiently merges dialect-specific models into a unified framework, promoting knowledge transfer and improving performance across dialects, making the system resource-efficient for real-world use.
3. SQL-GEN enhances execution accuracy by up to 20% over previous methods, making Text-to-SQL models more reliable for multi-dialect databases, with potential performance improvements in specific dialects like PostgreSQL and BigQuery.
4. By reducing the dependency on human annotations and enabling effective multi-dialect handling, SQL-GEN offers a cost-efficient approach to multi-dialect Text-to-SQL training. It would be better to open-source the generative model to help the community.

**Weaknesses:**

1. While SQL-GEN aims to create a unified model that bridges dialect gaps, merging dialect-specific models can dilute the depth of expertise for each dialect, leading to a possible trade-off between dialect-specific expertise and generalizability. It would be better to discuss the drawback of SQL-GEN in each dialect-specific example and provide a detailed analysis of performance on dialect-specific features or edge cases for each dialect. This could help clarify if and where any expertise dilution occurs.

2. The Mixture of Experts (MoE) approach enables the merging of dialect-specific models, but it also requires substantial computational resources for training, fine-tuning, and deploying models, especially as more dialects are added. It would be better to incorporate an easy SFT baseline to compare and provide a computational resource comparison between their MoE approach and a standard fine-tuning approach for multiple dialects. This would give readers a clearer picture of the trade-offs involved.

**Questions:**

1. How does SQL-GEN handle dialect-specific syntax that drastically changes query logic, such as proprietary SQL functions that are unique to one dialect? It would be better to provide specific examples of how SQL-GEN handles a few key proprietary functions from different dialects, and discuss any limitations in this area.
2. What measures are in place to ensure the synthetic queries generated by SQL-GEN align closely with real-world dialect usage beyond benchmarks?

---

> ### Author Response · Authors · 2024-11-21
> **Authors' response to reviewer**
>
> We sincerely thank the reviewer for their insightful comments and feedback, which have significantly enhanced the quality of our paper.
>
> > While SQL-GEN aims to create a unified model that bridges dialect gaps, merging dialect-specific ...
>
> We have included an analysis of the performance of the MoE models on each of the dialect-specific datasets, reported in Table 7. Interestingly, due to the larger capacity of the MoE model compared to the non-MoE expert models, and our proposed method of merging experts while preserving their feedforward layers, the MoE model's performance after fine-tuning did not degrade compared to fine-tuning on a single dialect (As shown in Table 7, we see roughly 3% gain). Compared with models trained on individual dialects (say CodeLlama 7B BigQuery expert in table 7) and a non-MOE model trained on mixture of the three data (CodeLlama 7B “generalist” in table 7) , our unified MOE model outperform they by 4% (31.14 vs 35.44), and 3% (32.83% vs 35.44) respectively. This improvement can partly be attributed to the considerable amount of shared knowledge across the dialect-specific datasets, as all tasks require common features, with only subtle differences stemming from dialect-specific features.
>
> > The Mixture of Experts (MoE) approach enables the merging of dialect-specific models, but ...
>
> Thank you for bringing this up, we actually conducted a similar experiment as you suggested, using SFT, which is reported in Table 7 under 'CodeLlama 7B generalist.' Specifically, we combined all the dialect-specific datasets and fine-tuned a CodeLlama 7B model to serve as our baseline all-dialect expert model. We compared this performance with our MoE-based approach, where our proposed method, even without fine-tuning, achieved similar performance to this baseline. After fine-tuning, our MoE model demonstrated an average improvement of 3% in accuracy across all dialects considered in the paper. Additionally, as per your suggestion, we have included a comparison of approximated memory consumption for inference, as reported below, considering our MoE model compared with deploying three different dialect expert models with 7B parameters. As demonstrated in the table below, our MoE model required much less GPU memory compared to keeping three different dialect specific models while keeping on par or better accuracy as reported in Table 7 of paper. Below we will provide an estimated GPU memory requirement for deploying our MoE model comparing to deploying three different expert models across different precisions:
>
> | Model                | float32   | bfloat16  | int8     | int4     |
> |----------------------|-----------|-----------|----------|----------|
> | MoE 3x7B            | 57.38 GB  | 28.68 GB  | 14.34 GB | 7.17 GB  |
> | 3 Dialect Expert 7B | 75.33 GB  | 37.68 GB  | 18.84 GB | 9.42 GB  |
>
> ### Questions:
>
> > How does SQL-GEN handle dialect-specific syntax ...
>
> To support dialect-specific functions, we provide a templated query along with a tutorial on how to use specific keywords or functions. We then ask the model to modify the input template query to incorporate the appropriate dialect-specific functions. Additionally, we have conducted a detailed analysis comparing the number of dialect-specific keywords used in various text-to-SQL baselines. Our approach demonstrates the highest number of samples utilizing dialect-specific keywords, as shown below:
>
> **keyword distribution figure**: https://anonymous.4open.science/r/SQLGEN-REEBUTTAL-02BD/keyword_distribution_analysis.png
>
> For instance, having the templated query “SELECT * FROM table” and the tutorial on how to use Postgres to_tsquery() function, our SQL-GEN method generated the following new template:
>
> ```sql
> SELECT * FROM table WHERE to_tsvector(column) @@ to_tsquery(literal);
> ```
>
> > What measures are in place to ensure the synthetic ...
>
> In order to make sure that the generated queries have the high quality we included both semantic and syntactic quality checking by using the existing dialect specific parser to make sure the queries are syntactically correct and also included semantic checks by using a LLM as a judge approach which is called quality checker model in the paper.

---

> > ### Comment · Reviewer_a2FH · 2024-11-21
> > **Thanks for the authors' efforts.**
> >
> > Regarding W1: While the unified MOE model performs better than the non-MOE model in terms of "results numbers", it still encounters errors—some of which the non-MOE model may handle better. Therefore, we are not only interested in the numerical improvements but also in gaining deeper insights through detailed error analysis and case studies.
> >
> > Regarding W2: It is expected that three single models together require less GPU memory compared to a single MOE model. I would like to see a comparison specifically against a single 7B CodeLlama generalist model for a more balanced evaluation.
> >
> > Regarding Q1: Grammatically correct queries are not necessarily meaningful. For instance, generated queries might include columns or values that are invalid in the real database context. In some cases, even gpt4 may struggle to determine whether the queries are executable on a real-world database.

---

> > > ### Author Response · Authors · 2024-11-26
> > > **Authors' response to reviewer**
> > >
> > > Thank you so much for your valuable follow up questions, here is our response to some of your concerns:
> > >
> > > > Regarding W1
> > >
> > > To address your concern regarding a detailed error analysis comparing the MoE model, the generalist 7B model, and models trained only on dialect-specific data, we evaluated the percentage of syntactically correct queries. Additionally, we calculated the percentage of queries correctly predicted by either the MoE model or the expert model, organizing the results into two categories:
> > >  1. **Queries with BQ keywords:** Queries requiring dialect-specific keywords
> > >  2. **Queries with General SQL knowledge:** Queries not requiring dialect-specific keywords and functions such as COUNT, WHERE, MAX, … which are supported by most of the dialects.
> > >
> > > A similar analysis was conducted for the generalist model versus the dialect-specific expert models. For the table below, the performance for Queries with BQ keywords and General SQL knowledge are relative to the BigQuery expert model so comparing the MoE model with BiQuery expert, MoE model solved 7.25% queries more than BigQuery expert while BigQuery expert solved 2.5% of queries with BigQuery specific keywords more than the MoE model.
> > >
> > > BigQuery (comparing MoE, comparing generalist 7B,  and BigQuery Expert):
> > >
> > > | Type of Queries                                   | MoE    | Generalist | BigQuery Expert |
> > > |--------------------------------------------------|--------|------------|-----------------|
> > > | Overall Performance on BigQuery                 | 32.25  | 32.25      | 27.5            |
> > > | Syntactically Correct Queries                   | 52.5%  | 47.5%      | 55%             |
> > > | Queries with BQ Keywords                        | 0%     | 2.5%       | 2.5%            |
> > > | Queries with General SQL Knowledge (PostgreSQL, SQLite, BigQuery) | 7.25% | 4.75% | 0.0%           |
> > >
> > > For the BigQuery dialect, as demonstrated above, comparing the MoE or generalist model with the BigQuery expert model reveals that the expert model possesses superior dialect-specific knowledge as expected. This is reflected in its higher percentage of syntactically correct queries and higher correctness for BigQuery-specific keywords. However, when it comes to general SQL knowledge, our MoE model outperforms the others (7.25% vs 4.75% or 0%, demonstrating that model merging effectively integrates dialect-specific expertise while maintaining competitive dialect-specific performance. Notably, the MoE model also achieves better performance in generating syntactically correct BigQuery queries compared to the generalist 7B model, highlighting the effectiveness of our proposed model-merging approach over simple fine-tuning on all available samples.
> > >
> > > > Regarding W2:
> > >
> > > The Total required memory footprint of the models for inference comparison between the MoE model and a single expert model as you requested:
> > >
> > > | Model       | float32   | bfloat16  | int8      | int4      |
> > > |-------------|-----------|-----------|-----------|-----------|
> > > | MoE 3x7B    | 57.38 GB  | 28.68 GB  | 14.34 GB  | 7.17 GB   |
> > > | Expert 7B   | 25.11 GB  | 12.56 GB  | 6.28 GB   | 3.14 GB   |
> > >
> > > However you should also consider the fact that the MoE model can serve multiple dialects, and to serve three dialects we have to deploy all three models and the required GPU memory is as reported before not the table above.

---

> > > > ### Comment · Reviewer_a2FH · 2024-11-27
> > > > **Thanks for resposnes**
> > > >
> > > > I believe the response Regarding W1 is excellent and could be integrated into the main text. However, concerning the second response, a model fine-tuned (SFT) on three domain datasets can handle three dialects, although its performance may degrade. Therefore, when considering the trade-off between GPU usage and performance comparison, MOE might not be the only viable option. Additionally, I would appreciate hearing the responses for Q1.

---

> > > > > ### Author Response · Authors · 2024-11-27
> > > > > **Authors' response to reviewer**
> > > > >
> > > > > Thank you so much for your valuable feedback on our response.
> > > > >
> > > > > > Regarding W1 is excellent and could be integrated into the main text
> > > > >
> > > > > For sure, we will include this experiment in the main paper as you suggested and thank you for pointing this out. We would greatly appreaciate it if you increase your assigned scores if we addressed your concerns.
> > > > >
> > > > > > trade-off between GPU usage and performance comparison
> > > > >
> > > > > Yes, as you suggested, there is a performance gap between the MoE model and the model fine-tuned on the three dialects. However, in low-resource settings, the small fine-tuned model (Not MoE) can be a better option. That said, quantization of the MoE model is also a viable approach, as int8 precision requires a smaller memory footprint compared to the fine-tuned model.
> > > > >
> > > > > >  responses for Q1.
> > > > >
> > > > > Thank you so much for bringing this up. For synthetically generated queries, there are two important factors that contribute to the construction of high-quality checks. The first is ensuring that queries are syntactically correct. This can be achieved either by using an SQL parser like SQLGlot or by actually executing the queries on a database. For example, works like the Gretel text-to-SQL dataset employed the first approach, but while these parsers can ensure syntactical correctness, they do not filter out queries that are syntactically valid but return empty results. In our work, we opted for the second approach—executing the queries directly on a database. This ensures that the queries are not only syntactically correct but also yield at least one row of results. This approach helps mitigate the issue you mentioned, where grammatically correct queries might lack actual meaning, as most of these meaningless queries typically return no results from the database. However, this method alone cannot address all cases of meaningless queries, making some form of semantic analysis necessary. To address this, we have two potential solutions: human annotation or verification using LLMs. While human annotation provides high accuracy, it is resource-intensive and costly, which is beyond the scope of this paper. Instead, we employed LLMs for verification purposes. Specifically, we designed a detailed prompt instructing the LLM to filter out queries that lack any business relevance. Furthermore, we supplemented the LLM's evaluation with external feedback by including the execution results of the synthetically generated queries. This enables the LLM to assess query quality based not only on the query itself but also on the returned results. Additionally, as discussed in Appendix A.4.1, we used a second LLM—different from the one used for query generation—for verification. This step helps reduce model-specific biases by leveraging the diverse knowledge of different LLMs.
> > > > >
> > > > > We hope the detailed answers and additional results provided below address your concerns. We kindly ask you to consider the possibility of a score adjustment

---

> > > > > > ### Comment · Reviewer_a2FH · 2024-11-27
> > > > > >
> > > > > > Thanks for the authors' efforts! We will have a discussion about adjusting the score.

---

### Official Review · Reviewer_VtYn · 2024-11-03

**Soundness:** 3
**Presentation:** 3
**Contribution:** 3
**Rating:** 6
**Confidence:** 3

**Summary:**

In the well-studied text-to-SQL task, the authors examine handling multiple SQL dialects and propose SQL-GEN, a three-step method to adapt text-to-SQL models trained on one SQL dialect to other dialects by generating synthetic text-to-SQL samples. They also introduce a Mixture of Experts approach to merge dialect-specific SQL models into a single model for efficient cross-dialect knowledge sharing and reducing maintenance costs by selectively activating dialect-specific experts based on the input. The method is tested on BIRD and Paglia benchmarks and is evaluated against baseline datasets to validate SQL-GEN's efficacy.

**Strengths:**

- **Originality**
    - The task of addressing dialects in text-to-SQL systems is nascently explored, and this paper makes a valuable contribution by introducing a novel MoE approach to manage multi-dialect scenarios effectively.
- **Quality and Clarity**:
     - The paper is generally easy to follow.
     - The experiments comprehensively test SQL-GEN across multiple benchmarks and datasets and validate the performance.

**Weaknesses:**

- Figures, especially Figures 3 and 4, are somewhat dense and challenging to interpret. A clearer visual representation or more detailed explanations would enhance accessibility.
- The text in Figure 5 could be enlarged to improve readability, as it currently appears difficult to see.

**Questions:**

- On a side note, it would be interesting to consider cross-lingual transfer methods for text-to-SQL as seen in multilingual NLP tasks, which might provide a useful baseline for comparison.

---

> ### Author Response · Authors · 2024-11-21
> **Authors' response to reviewer**
>
> We sincerely thank the reviewer for their valuable feedback. In response, in our updated paper, we provided additional explanations for Figures 3 and 4 to ensure the overall workflow presented in these figures is clear and understandable. Furthermore, as suggested, we have updated Figure 5 to enhance its readability, as shown below:
>
> **Updated keyword distribution figure**: https://anonymous.4open.science/r/SQLGEN-REEBUTTAL-02BD/keyword_distribution_analysis.png
>
> > On a side note, it would be interesting to consider cross-lingual transfer methods fo
>
> Thank you for the inspiring suggestions! Cross-lingual Text-to-SQL is indeed an exciting avenue for future exploration with our proposed methodology, and we greatly appreciate your insights. Our MoE model aligns with the philosophy of leveraging cross-dialect transfer between SQL variants to develop a multi-dialect-trained model, which demonstrates superior performance compared to models trained on a single dialect. This represents a step toward achieving effective “cross-lingual transfer methods for Text-to-SQL.” Additionally, training with large-scale, high-quality synthetic data offers significant potential to enhance transfer learning, enabling more data-efficient adaptation to new languages. By using our approach to generate diverse and comprehensive training samples, we aim to prepare models that generalize effectively across languages and SQL dialects, ultimately reducing the reliance on extensive annotated datasets for each new language.

---

### Official Review · Reviewer_GBQT · 2024-11-04

**Soundness:** 2
**Presentation:** 2
**Contribution:** 2
**Rating:** 5
**Confidence:** 4

**Summary:**

This paper introduces the SQL-GEN framework, which bridges the dialect gap for text-to-SQL systems by generating high-quality synthetic data for any dialect and merging dialect-specific models into a unified model using a novel Mixture of Experts (MoE) initialization method. The framework significantly improves execution accuracy on unseen real-world multi-dialect benchmarks and reduces the gap compared to large-scale human-annotated data.

**Strengths:**

1. The method is evaluated on three SQL dialects and proves its effectiveness.
2. The author has built a text-to-sql dataset that includes SQL dialects. If this dataset is open source, it will be of certain value to research.

**Weaknesses:**

1. The paper presents contributions that, while valuable, appear to have limited novelty compared to existing work in the field[1].
2. In addition, from the results, the improvement of synthetic data is not as good as that of bird train set.

[1] Yu T, Wu C S, Lin X V, et al. Grappa: Grammar-augmented pre-training for table semantic parsing[J]. arXiv preprint arXiv:2009.13845, 2020.

**Questions:**

No more questions. Please see above.

---

> ### Author Response · Authors · 2024-11-21
> **Authors' response to reviewer (Number one)**
>
> We would like to thank the reviewer for their valuable feedback that has helped us to improve our paper, we really appreciate it.
>
> **Novelty** We would like to emphasize our novel contributions: we introduced a tutorial-based method for generating synthetic Text-to-SQL datasets across multiple SQL dialects, marking the first approach of its kind and addressing a previously overlooked challenge in the Text-to-SQL domain. Additionally, we proposed a cutting-edge Mixture of Experts (MoE) approach, leveraging dialect-specific initialization to construct a unified model that integrates knowledge from various dialects. This approach not only outperforms dialect-specific models but also streamlines model deployment by consolidating multiple dialect-specific models into a single, efficient model.
>
> > The paper presents contributions that, while valuable, appear to have limited ...
>
> Next, we would like to explicitly compare our work with grammar-based approaches, such as the Grappa paper mentioned. Below, we highlight the key distinctions and contributions:
> 1. **Support for Multiple Dialects**: Our method is the first in the Text-to-SQL domain capable of generating synthetic samples for multiple SQL dialects in an automated way. It achieves this by leveraging dialect tutorials, enabling the generation of diverse question/SQL pairs that incorporate dialect-specific keywords and functions. In contrast, grammar-based approaches require access to pre-existing question/SQL pairs to induce a context-free grammar, which is often unavailable for many dialects.
> 2. **Flexibility Beyond Templates**: Grammar-based methods are inherently limited to specific grammars, restricting their ability to extend beyond predefined templates. Our method, however, generates arbitrarily complex queries, providing greater flexibility. This ability is crucial for avoiding overfitting to canonical query distributions, particularly when generating synthetic data to train LLMs.
> 3. **Automation and Lack of Human Supervision**: The Grappa method and similar approaches would necessitate manual annotation to replace entities in seed pairs with corresponding terminal types for each program template. In contrast, our approach is entirely automated, operating end-to-end without any human supervision.
> 4. **Sample Quality Assurance**: Grammar-based approaches often populate templates with randomly sampled columns and database values, leading to queries that may lack meaningful context or resemble natural, human-annotated samples. Our method mitigates this issue by incorporating a quality-checking model that filters out low-quality samples, ensuring the generated pairs provide meaningful business insights.
> 5. **Ease of Implementation and Adaptability**: Designing a context-free grammar is a non-trivial task, as noted in the Grappa paper, which identifies extending grammars as future work. Our method bypasses this limitation by fusing SQL templates with tutorial documents with LLMs, enabling the generation of sophisticated and contextually relevant queries without the need for predefined grammars.
> These differences underscore the novelty and practicality of our approach, setting it apart from grammar-based methods and offering significant advantages for generating high-quality, dialect-specific Text-to-SQL data that can be used to build multi-dialect Text-to-SQL systems.
>
> The second weakness is addressed in the next comment ....

---

> ### Author Response · Authors · 2024-11-21
> **Authors' response to reviewer (Number two)**
>
> >  improvement of synthetic data is not as good as ...
>
> Although models trained solely on our synthetic queries show slightly lower performance on the BIRD development set compared to those trained on the original BIRD training dataset, it is important to note that the BIRD development set aligns closely with the query distribution of the BIRD training set, inherently favoring models trained on the original BIRD data. More importantly,  to measure the value of synthetic data (not overfit on one particular distribution, like BIRD),  we evaluated the models on other datasets such as Pagila and the GitHub repository dataset. In these cases, models trained on our synthetic data outperformed those trained on the BIRD benchmark, with a large margin of roughly 20%, highlighting the effectiveness and generalizability of our approach for synthetic data generation across different datasets.
>
> As highlighted in Table 1, when evaluating on the out-of-domain benchmark Pagila, training with the BIRD dataset resulted in a -4.35% reduction in accuracy, whereas training on our dataset led to a 15.22% improvement. A similar trend was observed for the Github_repository dataset, where training on our dataset achieved a 10% improvement, while the BIRD dataset resulted in a -7.5% accuracy reduction.
>
> | Target Benchmark   | Train with BIRD (%) | Train with Our Synthetic (%) | Baseline (%) |
> |--------------------|----------------------|------------------------------|--------------|
> | Pagilla (CodeLlama 7) | 19.56 (-4.35)       | 39.13 (+15.22)               | 23.91        |
> | Github Repository   | 7.5 (-7.5)          | 25.0 (+10)                   | 15.0         |
>
> Furthermore, as demonstrated in Table 6, combining our synthetic examples with the original BIRD dataset improves performance on the BIRD dev set, beyond using the BIRD training set alone by 3-5% . This indicates that our synthetic dataset can enhance model capabilities even when high-quality training data is already available, as demonstrated below:
>
> | Training Dataset       | Model         | Execution Accuracy (%) |
> |-------------------------|---------------|-------------------------|
> | BIRD Train + Synthetic  | CodeLlama 7B  | **45.82**                  |
> | BIRD Train              | CodeLlama 7B  | 40.22                  |
> | BIRD Train + Synthetic  | CodeGemma 7B  | **51.10**                  |
> | BIRD Train              | CodeGemma 7B  | 45.63                  |
> | BIRD Train + Synthetic  | CodeLlama 7B  | **56.45**                  |
> | BIRD Train              | CodeLlama 7B  | 53.12                  |

---

> > ### Comment · Reviewer_GBQT · 2024-11-26
> >
> > Thank you for the detailed response, which addressed most of my concerns. Although your approach shows significant improvements on the Pagilla dataset compared to the Bird train set, it underperforms on BigQuery BIRD, PostgreSQL BIRD, and PostgreSQL Minidev when compared to models trained on the Bird train set. Additionally, Codellama demonstrates weaker code capabilities, and the generalizability of the dataset remains to be validated, such as  DeepSeek Coder、Qwen Coder.
> >
> > Therefore, I have decided to maintain my rating.

---

> > > ### Author Response · Authors · 2024-11-26
> > > **Authors' response to reviewer**
> > >
> > > Regarding the benchmarks, all the variants you mentioned—BigQuery BIRD, PostgreSQL BIRD, and PostgreSQL minidev BIRD—are derived from the original BIRD benchmark. PostgreSQL BIRD and BigQuery BIRD are simply translations of the SQLite queries into PostgreSQL or BigQuery dialects, maintaining a similar distribution as the original BIRD benchmark. For fair comparison to our datasets, we also translated the BIRD original training set into PostgreSQL and BigQuery for training purposes, ensuring the training set distribution has the same dialect as that of the testing set. The only two benchmarks that truly reflect the generalization capabilities of models trained on different datasets are PostgreSQL Pagila and BigQuery Github_repository. Our models, trained on our proposed datasets, significantly outperformed the models that are trained on the transpiled version of BIRD train set as demonstrated below: For instance training both CodeLlama 7B and Codestral 22B on our synthetic BigQuery dataset outperform the models trained on the BigQuery version of BIRD training set with 17.5 and 22.5 performance gap for CodeLlama 7B and Codestral 22B respectively.
> > >
> > > | Benchmark           | Model          | Transpiled BIRD Train Set | Our Synthetic Dataset |
> > > |---------------------|----------------|---------------------------|------------------------|
> > > | Github_repository   | CodeLlama 7B   | 7.5                       | 25                    |
> > > | Github_repository   | Codestral 22B  | 7.5                       | 30                    |
> > > | Pagila              | CodeLlama 7B   | 19.56                     | 39.13                 |
> > > | Pagila              | Codestral 22B  | 43.7                      | 50                    |
> > >
> > > Regarding your concern about the LLM, we should also mention that, in addition to the CodeLlama 7B model, we considered a much larger model—Codestral 22B—a recent code model with strong capabilities. As demonstrated above, the results from Codestral 22B were consistent with those of CodeLlama, with models trained on our dataset outperforming the BIRD benchmark on the Pagila and Github_repository benchmarks.

---

> > > > ### Comment · Reviewer_GBQT · 2024-12-02
> > > >
> > > > Thank you for the additional results.  After carefully reviewing the updates, I have decided to maintain my score.

---

> > > > > ### Author Response · Authors · 2024-12-04
> > > > > **Author's response to reviewer**
> > > > >
> > > > > Thank you for taking the time to review our paper and provide valuable feedback. Please let us know if there are any remaining concerns we can address.

---

### Meta-Review · Area_Chair_4UFk · 2024-12-21

**Metareview:**

This paper presents a method for generating synthetic data for text-to-SQL tasks in multiple "dialects" (e.g., BigQuery, PostgreSQL, SQLite). This is done by generating abstract queries which are then refined into each dialect.  Training on this data produces effective models for each setting. These models are then merged by initializing an MoE model with each expert using a novel procedure. Results across several benchmarks compare training on the synthetic data from this work to training on existing datasets such as BIRD, as well as the performance of the MoE model versus individual expert models.

This paper presents its problem setting very clearly, proposes a clear solution, and evaluates that solution very thoroughly. The mixture-of-experts approach is an interesting way to combine models.

The main weaknesses of the work are the model performance. Comparing to training on this paper's synthetic data to training on BIRD, and comparing MoE and SFT, the results are mixed (though generally positive in favor of this paper). The authors assert that their approach provides some conceptual advantages in both settings (e.g., generality compared to BIRD-trained models, where in-domain test sets are artificially easy). However, it's not clear to me whether the needle has been substantially moved in this domain (i.e., whether a broad range of practitioners in text-to-SQL really need to pay attention to these results), and also whether there's a generalizable takeaway method for a typical attendee of ICLR.

Overall, this paper delivers data and systems that will be useful to practitioners in this area, but I'm not sure these will be as useful or interesting to the broader ICLR audience.

**Additional Comments On Reviewer Discussion:**

GBQT brings up concerns about the model performance compared to training on BIRD. The authors rebut that the BIRD training data unfairly advantages models and they don't necessarily generalize. While I agree with this concern, I also think that synthetic data can be heavily engineered to a particular setting. I'm not sure the mixed results (wins on Pagilla and GitHub) are strong or convincing enough evidence here.

a2FH asks about comparing MoE and SFT. Although the MoE model performs better in some settings, it is larger than deploying separate models for separate settings, which is a valid concern in a production setting.

a2FH also asks about dialect-specific features and errors in each dialect. This point is somewhat addressed in the rebuttal.

---

### Decision · Program_Chairs · 2025-01-22

Reject